# Mechanical Properties and Durability of Rubberized and Glass Powder Modified Rubberized Concrete for Whitetopping Structures

**DOI:** 10.3390/ma14092321

**Published:** 2021-04-29

**Authors:** Audrius Grinys, Muthaiah Balamurugan, Algirdas Augonis, Ernestas Ivanauskas

**Affiliations:** Faculty of Civil Engineering and Architecture, Kaunas University of Technology, Studentų Str. 48, LT-51367 Kaunas, Lithuania; balamurugan.muthaiah@ktu.edu (M.B.); algirdas.augonis@ktu.lt (A.A.); ernestas.ivanauskas@ktu.lt (E.I.)

**Keywords:** rubberized concrete, freeze-thaw durability, porosity parameters, compressive strength, flexural strength, concrete fracture

## Abstract

This paper analyzes concrete fine aggregate (sand) modification by scrap tire rubber particles-fine crumb rubber (FCR) and coarse crumb rubber (CCR) of fraction 0/1 mm. Such rubberized concrete to get better bonding properties were modified by car-boxylated styrene butadiene rubber (SBR) latex and to gain the strength were modified by glass waste. The following tests—slump test, fresh concrete density, fresh concrete air content, compressive strength, flexural strength, fracture energy, freezing-thawing, porosity parameter, and scanning electron microscope—were conducted for rubberized concretes. From experiments, we can see that fresh concrete properties decreased when crumb rubber content has increased. Mostly it is related to crumb rubber (CR) lower specific gravity nature and higher fineness compared with changed fine aggregate-sand. In this research, we obtained a slight loss of compressive strength when CR was used in concrete However, these rubberized concretes with a small amount of rubber provided sufficient compressive strength results (greater than 50 MPa). Due to the pozzolanic reaction, we see that compressive strength results after 56 days in glass powder modified samples increased by 11–13% than 28 days com-pressive strengths, while at the same period control samples increased its compressive strength about 2.5%. Experiments have shown that the flexural strength of rubberized concrete with small amounts of CR increased by 3.4–15.8% compared to control mix, due the fact that rubber is an elastic material and it will absorb high energy and perform positive bending toughness. The test results indicated that CR can intercept the tensile stress in concrete and make the deformation more plastic. Fracturing of such conglomerate concrete is not brittle, there is no abrupt post-peak load drop and gradually continues after the maximum load is exceeded. Such concrete requires much higher fracture energy. It was obtained that FCR particles (lower than A300) will entrap more micropores content than coarse rubbers because due to their high specific area. Freezing-thawing results have confirmed that Kf values can be conveniently used to predict freeze-thaw resistance and durability of concrete. The test has shown that modification of concrete with 10 kg fine rubber waste will lead to similar mechanical and durability properties of concrete as was obtained in control concrete with 2 kg of prefabricated air bubbles.

## 1. Introduction

The road network is of vital importance for every country as part of their economic growth. In European Union countries, almost 90% of roads are of bituminous pavement. The maintenance costs of bituminous pavement is high compared to the construction costs [1]. The damaged asphalt layer can be covered by cement concrete instead of bituminous concrete and it is known as a whitetopping. Whitetopping is an effective rehabilitation solution for the damaged asphalt pavements. Whitetopping construction consists of two important techniques: (1) milling machine (used to remove the destructed asphalt layer’s thickness), (2) slip form paver (used to lay cement concrete) [2]. The initial cost of cement concrete is high compared to bituminous material. Still, cement concrete maintenance costs are less; moreover, cement concrete has higher strength and durability. The main advantage of cement concrete is its high albedo value compared to bituminous concrete. Whitetopping construction is particularly suitable when the existing roadway structure is no longer enough due to high static and dynamic traffic loads. Usually, a milled layer will replace with prefabricated air burbles (which act as an air-entraining agent and will provide high resistance to freezing and thawing) ([3]), low shrinkage, polypropylene macro fiber-modified high-performance concrete to increase traffic areas’ load-bearing capacity. After milling, asphalt layer thickness should be at least 8 cm, and the thickness of the new cement concrete layer should not be less than 10 cm. The above-described thickness values are given in the form of theory. In reality, the thickness of cement concrete pavement depends on traffic loads, environmental conditions, and previous asphalt stages. Whitetopping is classified into three types: Conventional whitetopping (thickness more than 200 mm), thin whitetopping (thickness from 100 mm to 200 mm), and ultra-thin whitetopping (thickness from 50 mm to 100 mm). 

There is a massive amount of different waste products getting disposed of every year all over the world and some of the wastes can be used in whitetopping concrete [4]. Among wastes, rubber tires, waste glass, and fly ash are concentrated on in this research. Worldwide automobile manufacturing increases every year, due to this, the production of tires is also rising. Vehicle tires are made from chemicals. By disposal, chemicals will become toxic to the environment. Years ago, the rubber wastes were usually dumped into the land, stockpiled in the industry, and disposed of by burning, which belongs to environmental decline. As a result of waste disposal, humans faced many problems such as fire cause and reproduction of mosquitoes and rats in stockpiled areas. In the European Union, tire production from 2010 to 2018 is 4.5 million tonnes to 5.1 million tonnes. According to the directive disposal of waste 1000/31/EC, European countries banned disposal and stockpile of whole tires from July 2003; from July 2006 they banned ground rubber disposal. The end of life vehicle directive 2000/53/EC introduced three acts of legislation to improve waste tire management: they are extended producer responsibility, a tax system, and the free market system. According to this legislation, waste rubber tires started to be resused all over European countries, and disposal rates also started reduced. The European tire and rubber manufacturers association (ETRMA) is managing end-life tires data; up to 2017, 92% of (ELTs) were collected and recycled [5]. The uses of waste rubber tires: making plastic and rubber products, fuels for cement kiln, and base layer for asphalt pavement. Naturally, rubber will provide toughness, impact resistance, and freezing-thawing effect in concrete [6,7,8,9,10,11,12,13,14,15,16,17,18,19].

Rubber is a naturally hydrophobic material. Generally, a high amount of silicon contributes to rubber’s hydrophobic nature, and in some cases zinc stearate also creates this hydrophobic nature (zinc stearate is one of the tire manufacturing product) [20]. Rubber’s hydrophobic nature can be eliminated by treating with adhesive material. An adhesive material can be liquid or solid; it is used to create bonding between two dissimilar materials. Generally, polymers are used as adhesive material in concrete. The coupling agents are also creating a better bond between two dissimilar materials. In this research, we planned to use SBR latex to bring a good bond between rubber and cement paste. The secondary advantage of polymers is keeping rubber particles stable and avoiding agglomeration during vibrator compaction. To improve the interfacial compatibility some authors [21] treated tire rubber with oxidant reactives in order to create polar groups on the surface which would improve the compatibility with other materials.

In the world, about 130 million tonnes of glasses are generated annually [22]. In 2018, the European Union’s glass production reached about 36.5 million tonnes, which is slightly higher than before the year 2017. Therefore, year by year, glass production is increasing, and in this way, glass wastes are also being increased. Europe is one of the largest glass producers globally compared to China and North America. Only a small fraction of the solid wastes are recycled directly to the primary market, i.e., the bottling and container industry. The remaining glass wastes are discarded into the land. Glass is inert material; it will not decompose and remains in the land for many years. The disposed glass will affect the land quality and water table. Hazardous glasses such as cathode ray tubes and fluorescent lamps are even more high risk of affecting the land. According to AASHTO, waste glass absorbing a high load when substituting as a base layer for pavement, and glass providing good results than conventional asphalt. Benefits of glass powder wastes in concrete are as follows: increases durability, low shrinkage, high abrasion, and low water absorption [23,24,25,26].

In European countries, pavement structures are mainly affected by the freezing-thawing effect, which reducing concretes age. Air-entraining agents can control a freezing-thawing effect. The researchers found that waste crumb rubber is acting similar to a traditional air-entraining agent because rubber entrapping air in concrete due to its non-polar surface nature—entrapped air creating pores called airvoids. Those pores help to hold and release the water pressure and protects from a freezing-thawing effect. Pore size depends on aggregates, and pore spacing should be around 0.25 mm for better freeze-thaw.

By continuing previous research [3] the main aim of this research work is to analyse the effect of crumb rubber used as fine elastic aggregate on the mechanical and durability properties of concrete and find the lowest amounts of rubber where concrete fulfills durability requirements but strength properties will not lose or will be similar compared to ordinary concrete for whitetopping structures. Also to get better strength properties rubberized concrete was modyfied by glass powder wastes. In this research strength properties, fracture energy, freeze-thaw resistance, water absorption by immersion, porosity parameters, and analyzed microstructure of concrete was investigated.

## 2. Materials

Concrete mixes with different fineness crumb rubber from different suppliers (Figure 1), glass powder wastes, SBR latex and prefabricated air bubbles (Sika Aer Solid) with the same water and cement ratio (W/C) were prepared for this experimental research. The water amount in SBR latex was calculated into W/C ratio. Ordinary Portland cement CEM I 42.5 R of the fineness 410 m^2^/kg was used. The chemical composition of cement is given in Table 1. 153 L of water were used to produce the cement slurry of normal consistency. Sand of fraction 0/4mm, particle density 2650 kg/m^3^, was used as fine aggregate. A portion of sand was replaced by crumb rubber obtained from used tires. Crushed granite of fraction 4/16mm, particle density 2720 kg/m^3^ was used as coarse aggregate. All concrete mixes were made with the same amount of coarse aggregate, 999 kg/m^3^ of concrete. Polycarboxylate polymer-based plasticizing admixture was used. Modified polycarboxylates properties: appearance–light brown liquid, density—1.07 ± 0.005 kg/lit, pH value—4.5 ± 1, chlorine ion content was <0.2% by weight, sodium oxide content was <0.4% by weight. In this research, 0.8% (wt. of cement) of water-reducing admixture were used in concrete. Organic compounds-based shrinkage reducing admixture was used in this research. Shrinkage reducing admixture properties: appearance–transparent liquid, density-0.94 ± 0.02 kg/lit, pH value-6.0 ± 1, chlorine ion content was <0.1% by weight, sodium oxide content was <0.1% by weight. In this research, 2.0% (wt. of cement) of shrinkage reducing admixture were used in concrete. Coarse crumb rubber 0/1fr. (CCR) and fine crumb rubber 0/1fr. (FCR) was obtained from the different local waste tire recycling companies. Rubber particle size distribution (fine crumb rubber and coarse crumb rubber) is shown in Figure 1. From the figure, we can clearly understand that fine crumb rubber is much finer and has around 20% 0.25 mm particles (more A300 size according to EN 480-11) compared with coarse crumb rubber which has only 0.5% 0.25 mm particles. Rubber particle density varies 1010–1020 kg/m^3^, bulk density 475–485 kg/m^3^. CR was added 5 kg/m^3^, 10 kg/m^3^ and 20 kg/m^3^ in concrete and part of sand was changed by CR. In this research, mixed white and green color waste beverage bottles were crushed with an electronic crushing machine. According to ASTM C618-02, recycled glass powder is an excellent pozzolanic material. In this research, we used glass powder with size <300 μm. Properties of glass powder: density—2266 kg/m^3^, bulk density—1245 kg/m^3^. The chemical composition of glass powder is given in Table 1. Here we can see that glass powder mainly consists of SiO_2_ and Na_2_Oeq oxides. Acrylonitrile polymer-based Sika Aer Solid was used as pre-fabricated air bubbles. The density of prefabricated air bubbles is 200 kg/m^3^. In this research, liquid-polymer-based carboxylated styrene butadiene latex was used for rubber surface treatment. Crumb rubber was dispersed entirely in SBR latex using the ultrasonic dispersion method. Time taken for dispersion about 1min and this process was carried out at power 250 W. Properties of SBR latex: density ~1.03 kg/dm^3^, pH value ~10. The fiber used in this research is chemically based on polyolefine. Polyolefine fiber comprises 85% of polypropylene, and polyolefine is from the polypropylene and polyethylene family. Properties of polyolefine fiber: density ~0.91 kg/L, melting point ~170 °C, tensile strength ~430 MPa, and tensile modulus of elasticity ~6 GPa. Polyolefine fiber dimensions: 60 mm length and 0.84 mm diameter.

## 3. Experimental Procedure

The concrete mixes were prepared in the laboratory using a forced type Zyklos concrete mixer. The concrete was mixed and the concrete specimens were formed according to standard EN 206.

In this study, crumb rubber granulometry was done according to EN 933-1 (Figure 1), the air content of compacted fresh concrete was determined according to standard EN 12350-7, the slump according to standard EN 12350-2, the density of hardened concrete specimens according to standard EN 12390-6, the compressive strength of hardened concrete according to standard EN 12390-3, the flexural strength test was carried out according to EN 12390-5, and the freeze-thaw resistance according to standard LST L 1428.17. The porosity parameters of investigated concrete series were determined by measuring the kinetics of water absorption according to the previous procedure [3,27]. Fracture energy was calculated from CMOD (crack mouth opening displacement) curves [28]. The Originpro software is used in this research to find an area under the CMOD curve. The following formula is used to calculate fracture energy.
(1)$$G_{F}=\frac{W_{t}\;}{\left(D-a_{o}\right)b}$$
where:

*G_F_* is fracture energy, *W_t_* is total energy, *D* is specimen depth, *a_o_* is notch depth, *b* is the width of the specimen,
(2)$$W_{t}=W_{r}+2P_{W}\delta f.$$
where:

*W_r_* is the area under CMOD curve, *P_w_* is equivalent self-weight force, *δf* is displacement under the curve,
(3)$$P_{W}=\frac{W_{o\;}S}{2L}\;$$
where:

*W_o_* is the weight of the specimen, *S* is span length, *L* is the length of the specimen.

## 4. Results and Discussion

Different mixes were made under laboratory conditions to determine the effect of crumb rubber addition on the durability properties of hardened concrete: reference mix with prefabricated air bubbles (Control), concrete with different fineness and amounts of crumb rubber, rubberized concrete with SBR latex and rubberized SBR latex modified concrete with glass powder wastes were prepared (Table 2) for the study.

### 4.1. Fresh Concrete Test Results

Fresh concrete properties are described in Table 3. Here we can see that the workability of concretes with different materials are different. All batches with a small amount of CCR have a higher slump value than the control mix, but the FCR were showed less slump value than a control mix. A control mix slump value was 195 mm, which comes under the S4 slump class (very high workability). Overall, when rubber content increased, the slump value gets decreased due to its irregular shape and fineness. The same performance has been noticed in this literature [10] explained that slump value decreased due to no free water in concrete because fine rubber absorbed more water than sand. However, in this research, we believe that fine crumb rubber acts as a filler in concrete. Therefore, it occupied more pores when a rubber amount increased, which made concrete compact and reduced the fresh concrete slump. Rubberized concrete mix modified with SBR latex showed higher slump values compared to control mix due to the fluid nature of SBR latex and higher porosity which gives more softer concrete mix. When adding glass powder, SBR latex and CCR, the slump value was got the highest values. The reason behind this high workability: (1) The water for hydration becomes free water due to the reduction of cement (glass replacement); that free water made fresh concrete high flowable, (2) Due to the smooth surface of a glass particle (the surface nature of glass described in SEM analysis), (3) The superplasticizer were reduced glass surface tension and made a glass concrete flowable. The same result with this explanation can see in this literature‘s [24,26,29,30].

The fresh concrete density and air content are interconnected. When air content gets increased, fresh concrete density will get decreased. Changing materials in concrete can change their property. In the fresh concrete density test, we can see that density varies for different materials. The fresh concrete density for a control mix is 2450 kg/m^3^. Naturally, rubber was a low specific gravity material than the fine aggregate. From experiments we can see that when crumb rubber content increased, fresh concrete density decreased due to its low specific gravity nature. There is another important reason for density lowering: air content (entrapped by rubber–nonpolar nature) rises due to the high specific area of fine crumb rubber. Adding glass powder with treated rubber were showed lower density than other samples because both glass powder and rubber have a very low particle density than sand and cement. Therefore, glass rubber concrete’s fresh density will decrease when cement (glass replacement) and sand (rubber replacement) content are reduced, these all are basic reasons. Additionally, there is another compelling reason for lowering glass concrete fresh density: glass particles are also entrapping air in concrete due to its surface nature and geometry. As a result, increasing rubber and glass content, air voids will increase, and fresh concrete density will decrease.

The air in concrete is essential for freezing-thawing resistance. We used a special air-entraining agent (prefabricated bubbles) in a control mix, which executed about 2.5% air content. The fresh concrete pores are almost closed porosity. The rubbers’ non-polar nature will repel water and entraps air easily into the concrete. From Table 2 we see that when crumb rubber increased, fresh concrete air content increased due to the high specific area of fine crumb rubber. Therefore, by increasing CR, more air will get entrapped into the concrete was explained by [7]. When CR modified concrete was mixed with SBR latex air content increased more. However, the high content of SBR can be a secondary reason for high air content. The rubber glass fresh concretes exhibited highest air content than all other samples. In GLCR20 mix, when increased glass powder and treated rubber amount, air content increased. High air content in the GLCR10 and GLCR20 batches was due to glass particle irregular shape and sharp edge, and also due to high specific area of fine glass powder that entraps more air when glass amount increases. The following research [31] explains that glass particles were entrapping air because of their irregular shape, and importantly, sharp edges of glass particles were carrying air into the concrete.

### 4.2. Hardened Concrete Test Results

#### 4.2.1. Strength Properties and Fracture

The compressive strength of the control mix is 56.8 N/mm^2^ (Figure 2). From this literature point [14], fine crumb rubbers are more effective in concrete mechanical field than coarser rubber particles. Generally, substituting fine crumb rubber as sand instead of cement providing good compressive performance. When crumb rubbers were substituted as a fine aggregate in concrete, compressive strengths reduce for 5, 10, and 20 kg of CCR and 10 kg of FCR was about 7.04%, 11.79%, 11.97% and 3.87% than a control mix. When rubber amounts increased, compressive strength gets decreased due to rises of air voids and cracks (which will develop easily around soft rubber materials). However, these rubberized concretes provided sufficient compressive strength results (greater than 50 N/mm^2^). The efficient compressive strength of CR concretes was achieved due to tiny size rubber particles occupied micropores between cement paste and aggregates and because of small amounts of CR was added in concrete. The literature [32] proved that fine crumb rubber concrete’s compressive strength was nearer to the control mix.

Crumb rubber has been treated with SBR latex to ensure homogenous rubber distribution and better bonding between rubber and cement paste. SBR latex will form a chemical bond between rubber and cement hydration products (C-S-H) for superior strength. In this research, we can see that SBR latex modified rubber concrete showed less compressive strength than rubberized concrete without SBR latex and control mix. Strength reduces for 5, 10, and 20 kg of LCR was about 21.1%, 16.9%, and 17.9% compared with control samples average compressive strength values. The reason behind insufficient compressive strength is due to the high amount of SBR latex, which brings more porosity to concrete (porous nature of crumb rubber samples is shown below SEM pictures). That porosity made concrete less dense and weakened under compressive force. The literature [33] showed rubber treated with SBR latex and silane coupling agent executes better compressive strength than untreated rubber concrete and control mix. The additional supporting bond (like silane coupling agent) making concrete strength high. Not used a supporting adhesive could be a secondary reason for strength loss in this research.

Compressive strength of rubberized concrete with glass powder waste was tested after 28 days and 56 days because the pozzolanic reactions of glass powder will take place at later stages. Glass powder waste modified concrete compressive strength results after 28 days: GLCR10—45.1 N/mm^2^, GLCR20—44.1 N/mm^2^. Compressive strength reduced after 28 days for batches GLCR10, GLCR20 about 20.6%, 22.3% than a control mix. Compressive strength results of samples where glass powder was added after 56 days was: GLCR10—50.1 N/mm^2^, GLCR20—49.8 N/mm^2^. Strength reduced after 56 days for batches GLCR10, GLCR20 about 13.9%, 14.4% than a control mix at the same age. We see that compressive strength results after 56 days in glass powder modified samples increased 11–13% than 28 days compressive strengths, while in control samples at the same period was obtained 2.5% compressive strength increase. Therefore, from these 56 day results, we can say that pozzolanic reactions of glass powder started working in rubberized concrete. However, we can expect that glass powder will increase concretes strength slowly and at later stages due this pozzolanic reaction. Elaqra et al. [26] used fine glass powder as cement, which showed greater compressive strength than the control mix after 90 days. The same results were obtained in this literature [34] with glass powder and fine rubber aggregate achieved their strength greater than the control mix after 90 days.

In this research, a three-point loading method was used for finding flexural strength. A control mix flexural strength was obtained 8.48 N/mm^2^ (Figure 3). The significant flexural strength of a control mix was achieved due to the efficiency of polyolefine fibers. Fibers in each layer made concrete to withstand the load. Naturally, rubber is an elastic material; it will absorb high energy and perform positive bending toughness. The flexural strength for CCR5, CCR10, and FCR10 increased by 15.8%, 5.7%, and 3.4% compared with a control mix. The flexural strength of CR20 was reduced slightly by 1.2% than a control mix (but strength was nearer to control specimen). Fine crumb rubbers filled the pores in concrete, which reduced the stress development at the pores, leading to higher flexural strength for these samples.

The flexural strength for LCR5, LCR10, LCR20, GLCR10, and GLCR20 were reduced by 21–32%, compared with a control mix. The reason behind this failure: (1) Much lower compressive strength results which is related with flexural strength results; (2) A high amount of SBR latex liquid used in concrete, which developed more porosity; (3) There was no strong bond between rubber and cement paste (need additional adhesive promotion). The literature [33] explained that rubber with an additional coupling agent and SBR latex improved flexural strength than the untreated crumb rubber concrete and control mix. The same literature said that a high number of polymers could lead to strength loss. Adding glass powder with treated rubber has reduced flexural strength during the initial stages. However, we can expect better strength at later stages because the pozzolanic reaction of glass powder will activate during concretes later ages.

Concrete potential against fracture can be determined by calculating fracture energy. In this research, fracture energy was calculated to estimate crumb rubber toughness and fibers toughness. Fracture energy can be calculated by finding an area under a flexural stress-strain curve until failure. An area under the curve tells about the ability of concrete energy absorption. For example, a larger area represents that concrete can absorb greater energy before failure. Generally, fiber-reinforced concrete will take a long time for failure than non-fiber-reinforced concrete, and fiber-reinforced concrete will have more significant displacement and area (under the CMOD curve). In this research, polyolefine fiber was used in all samples, and we used Originpro analyzing software to find an area under CMOD curve (Figure 4a–c). From Table 4, we can see the calculated area for respective samples. A control mix calculated fracture energy is 973 N/m. Higher fracture energy was obtained in samples CR5 and CR10 (1222 N/m, 1161 N/m), while similar fracture energy values was obtained in samples CR20 and FCR10 (883 N/m, 954 N/m) compared to control samples. These test results indicated that CR can intercept the tensile stress in concrete and make the deformation more plastic. Fracturing of such conglomerate concrete is not brittle, there is no abrupt post-peak load drop and it gradually continues after the maximum load is exceeded. Such concrete requires much higher fracture energy. The same result was found in this literature [15], where fine crumb rubber concrete had a higher fracture energy than the control mix. Also from Table 4 we can see that all samples except LCR20 achieved residual flexural strength values of 1.5 MPa at 0.5 mm CMOD and a residual flexural strength of 1MPa at 3.5 mm CMOD which is described in EN 14889-2 and it is normative for fiber-reinforced concretes.

#### 4.2.2. The Effect of Crumbed Rubber on Freeze-Thaw Resistance of Concrete

In this research, we investigated the performance of crumb rubber, SBR latex modified crumb rubber and crumb rubber with glass powder concretes after 200 freezing-thawing cycles. The freezing-thawing results of all samples were compared with control concrete. Generally, there should be a minimum amount of air in concrete to perform against the freeze-thaw effect. A traditional air-entraining agent or special air-entraining agent will use in concrete for a better freeze-thaw effect in the industries. So many years before, researchers found that rubber can act as an air-entraining agent in concrete. Due to its non-polar nature, rubber entraps air in concrete, which provides space for pressure release during water freezing-thawing. Fine crumb rubbers will entrap more air content than coarse rubbers because due to their high specific area. A rubber amount should be reasonable for sufficient air content in concrete. In this research, we added 5, 10, 20 kgs of crumb rubbers in concerts. After 200 cycles, a control mix compressive strength was obtained 57.5 MPa; it increased by 1.19% compared to zero cycles compressive strength (Figure 5). After 200 freezing-thawing cycles, compressive strength for CR5, CR10, and CR20 decreased up to 36.13% (33.7 MPa), 64.17% (18.0 MPa), and 56.62% (21.7 MPa) from before freezing-thawing cycles compressive strength values. This could be explained by not enough amount of fine rubber particles (lower than A_300_) which positive influence resistance to freezing-thawing of concrete. While SBR latex modified rubberized concrete compressive strength after 200 freezing-thawing cycles increased by 11.98% (50.2 MPa), 8.59% (51.1 MPa) and 8.33% (51.1 MPa) for LCR5, LCR10, and LCR20 samples accordingly than pre-freeze-thaw compressive strength values. In LCR samples was obtained a high amount of porosity which gave the strength reduction, but the same high amount of porosity gave the positive durable property to concrete. We can see the porosity nature (due to SBR) of LCR set concrete from the microscopic analysis. Those porosities gave space for water expansion due to freezing-thawing. Freezing-thawing resistance of concrete modified by glass powder waste are shown in Figure 5. Here we can see that after 200 freezing-thawing cycles, batches with GLCR10 and GLCR20 compressive strengths increased up to 10.04% (49.6 MPa) and 7.87% (47.6 MPa) compared with strength results before freezing-thawing cycles. We believe that due pozzolanic reactions of glass powder it was filled the pores and reduced the amount of open porosity of concrete; due to the fact that GLCR batches were performed well during freezing-thawing cycles. The pozzolanic reaction of glass powder are shown in the SEM analysis part, and the porosity nature discussed in the water absorption kinetics part. Tests have showed that concrete with the finer particles of crumb rubber withstands the freeze-thaw effect after 200 cycles. Pre-freeze-thaw compressive strength of FCR10 is 54.6 MPa. After 200 cycles, it increased by 1.98% (55.7 MPa) than pre-freeze-thaw compressive strength. From rubber particle size distribution (Figure 1), we can see that fine crumb rubber 0/1fr. passing through 0.25mm (lower than A_300_) sieve is 20%, while in coarse crumb rubber is 1%. We can state that fine size crumb rubber created more micropores, which made concrete more durable during the freezing-thawing resistance.

Porosity is the prime factor for concrete performance in a mechanical and durable environment. Concrete’s total porosity, open porosity, and close porosity are calculated by using a water absorption test. After 48 h of water absorption test, a control mix absorbed 4.03% of water (Table 5). Untreated crumb rubber concretes absorbed little more water than a control mix. In that, CR10 concrete showed less water absorption than a control, CR5, and CR20 concretes. The reason behind less water being absorbed by CR10 concrete is that it consists of less open porosity and more close porosity than a control mix and the other two batches. The amount of water absorbed by CR5, CR10, and CR20 is about 4.11%, 3.72%, and 4.11%, respectively. In this literature [32], the water absorption rate is increased along with increased rubber content. After that the water absorption rate was slightly increased in SBR latex modified rubberized concretes and FCR10 due to more open porosity. The amount of water absorbed by LCR5, LCR10, LCR20 and FCR10 is about 4.46%, 4.47%, 4.48% and 4.35%, respectively.

When glass powder was added to SBR latex modified rubberized concretes, it decreased the amount of open porosity and increased close porosity content (Figure 6). Due to less open porosity, the water absorption rate decreased in rubber glass concrete. The amount of water absorbed by GLCR10 and GLCR20 is about 3.27% and 3.26% (Table 5). Here, we notice that glass powder rubberized concrete showed the least water absorption rate than all other batches. This literature [26] also confirmed that glass’s pozzolanic reaction reduces the water absorption rate in concrete.

From Table 5, we can see that SBR latex modified rubberized concretes and concrete modified with fine rubber samples are with a high frost resistant factor (Kf) than all other samples (LCR—7.93, LCR10—8.97, LCR20—9.26, GLCR10—18.16, GLCR20—19.97, FCR10—5.52). According to the high frost resistant factor and high predicted cycles, compressive strength for these samples are increased after 200 cycles. Therefore, from the Kf factor, we can predict the performance of concrete in freezing-thawing [3]. The least frost resistant factors are obtained by CR5—1.55, CR10—2.16, CR20—2.18. According to the least frost resistant factor, samples (5UCR, 20UCR, 10UFCR) were performed worst in a freezing-thawing test and was not keeping 200 freezing-thawing cycles. A control mix also obtained a low frost resistant factor, but it performed well after freezing-thawing cycles. The reason behind control concrete performance is that prefabricated air bubbles contain lots of microbubbles, which protected a control concrete under the freezing-thawing effect. The same result, low frost resistant factor, and highly durable prefabricated air bubble concrete are shown in this literature [16]. Also we can see that concrete modified with 10 kg/m^3^ of fine crumb rubber showed less frost resistant factor (Kf) compared with LCR and GLCR mixes but it exhibited good freezing-thawing performance. The reason behind that performance is that fine crumb rubber is a very finer particle than sieved normal crumb rubber, so FCR10 entrapped more micropores in concrete (A_300_), which made good durable performance in the freezing-thawing test.

#### 4.2.3. Scanning Electron Microscopic Analysis

In this paper, a scanning electron microscope was used to project: (1) prefabricated air bubbles in control concrete (Figure 7a); (2) crumb rubber geometry and its bonding with cement stone (Figure 7b); (3) SBR latex modified crumb rubber bonding with cement stone analyses (Figure 7c); (4) pozzolanic activity of glass powder in rubberized concrete (Figure 7d); (5) fine crumb rubber contact zone with cement stone analyses (Figure 7e). From Figure 7a we can see that prefabricated air bubbles in concrete structure was obtained in different sizes (fine and coarse). We can see that prefabricated air bubbles are spherical structure and it is evenly dispersed in concrete structure provided. Thus, gave enough space for water to release its pressure during the freezing-thawing effect, due to this reason control concrete was performed well after freezing-thawing cycles. Figure 7b shows contact zone between crumb rubber and cement stone. Here we can see that crumb rubber is irregular in shape. Due to its irregular shape, there is a greater chance that it can entrap more air in cement stone-crumb rubber contact zone, but also we can see that these pores are big and can be described as compaction or cavern pores which gives negative effect due concrete freezing-thawing resistance. Also due irregular size of crumb rubber we got better bending with cement stone and better concrete fracture parameters.

SEM analysis clearly showed that SBR latex modified crumb rubber concrete contains lots of pores; those pores decreased such concrete performance in both the compressive and flexural field. However, the pores helped positively during the freezing-thawing effect. Porosity parameter analysis also explained that SBR latex modified rubberized concrete has high porosity amount than other investigated concretes untreated crumb rubber concrete. Figure 7d shows that glass waste particles have a smooth surface and sharp edges. Due to its smooth surface nature, fresh concrete slump value gets increased for rubber glass batches. Its sharp edges carried air in fresh concrete, which increased air content for rubber glass fresh concretes. SEM image of FCR particles the same as CR particles shows the irregular shape of particle. However, here we can see that FCR particles is much smaller that CR and it gives smaller pore size around rubber and cement stone contact zone. These pores give better damping effect for freezing water and gives better concrete resistance to freezing-thawing.

## 5. Conclusions

The following conclusions are made in this research:When rubber content increased, the workability decreased due to its irregular shape and fineness and that fine crumb rubber acts as a filler in concrete. Rubberized concrete mix modified with SBR latex showed higher workability compared to control mix due to the fluid nature of SBR latex and higher porosity which gives more softer concrete mix. When glass powder was added, it increased workability due to glass particles smooth surfaces.From experiments we can see that when crumb rubber content increased, fresh concrete density decreased, and air content increased due to its low specific gravity nature. Adding glass powder in rubberized concrete were showed lower density than other samples because both glass powder and rubber have a very low particle density than sand and cement.When rubber amounts increased, compressive strength get decreased due to rises of air voids and cracks (which will develop easily around soft rubber materials). However, these rubberized concretes with a small amount of rubber provided sufficient compressive strength results (greater than 50 MPa). We see that compressive strength results after 56 days in glass powder modified samples increased 11–13% than 28 days compressive strengths, while in control samples at the same period was obtained 2.5% compressive strength increase. Therefore, from these 56 day results, we can say that pozzolanic reactions of glass powder started working in rubberized concrete.The flexural strength of rubberized concrete with small amounts CR were increased by 3.4–15.8% compared with a control mix, due the fact that rubber is an elastic material and it will absorb high energy and perform positive bending toughness. The test results indicated that CR can intercept the tensile stress in concrete and make the deformation more plastic. The fracturing of such conglomerate concrete is not brittle, there is no abrupt post-peak load drop and gradually continues after the maximum load is exceeded. Such concrete requires much higher fracture energy.Due to its non-polar nature, rubber entraps air in concrete, which provides space for pressure release during water freezing-thawing. Fine crumb rubber particles (lower than A300) will entrap more air content than coarse rubbers because due to their high specific area. We can state that 10 kg/m^3^ of fine size crumb rubber created enough micropores, which made concrete durable during the freezing-thawing resistance. Freezing-thawing results have confirmed that Kf values can be conveniently used to predict freeze-thaw resistance and durability of concrete.In SEM analysis we can see that fine crumb rubber particles are much smaller that CR and it gives smaller pore size around rubber and cement stone contact zone. These pores and rubber particles give damping effect for freezing water which gives better concrete resistance to freezing-thawing.From all results we can state that 2 kg/m^3^ of prefabricated air burbles can be successfully replaced by 10 kg/m^3^ of fine crumb rubber to get the similar mechanical and durability properties.

## Figures and Tables

**Figure 1 materials-14-02321-f001:**
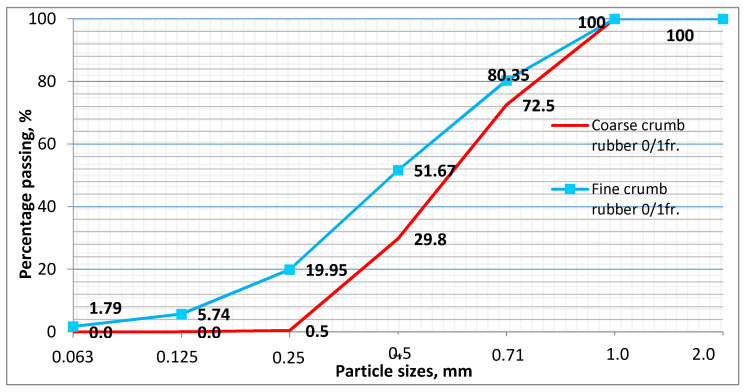
This is a figure. Schemes follow the same formatting.

**Figure 2 materials-14-02321-f002:**
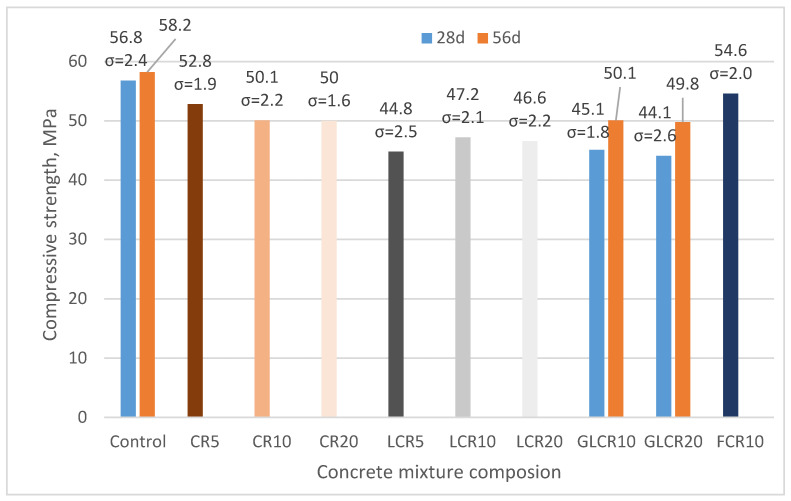
The change in compressive strength of concrete.

**Figure 3 materials-14-02321-f003:**
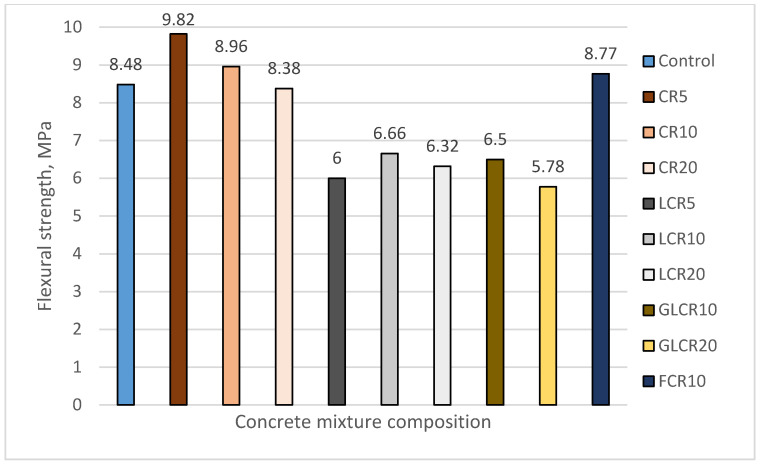
The change in flexural strength of concrete.

**Figure 4 materials-14-02321-f004:**
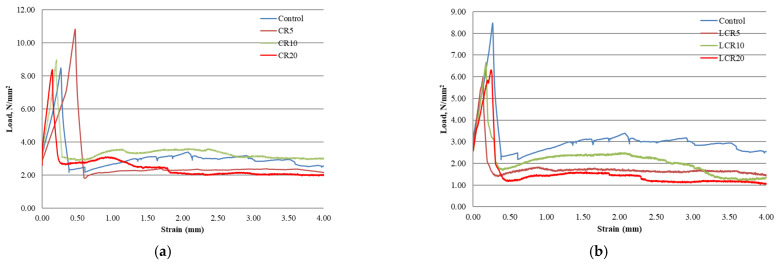
The function of stress and CMOD of concrete with CR (**a**), LCR (**b**) and GLCR&FCR (**c**).

**Figure 5 materials-14-02321-f005:**
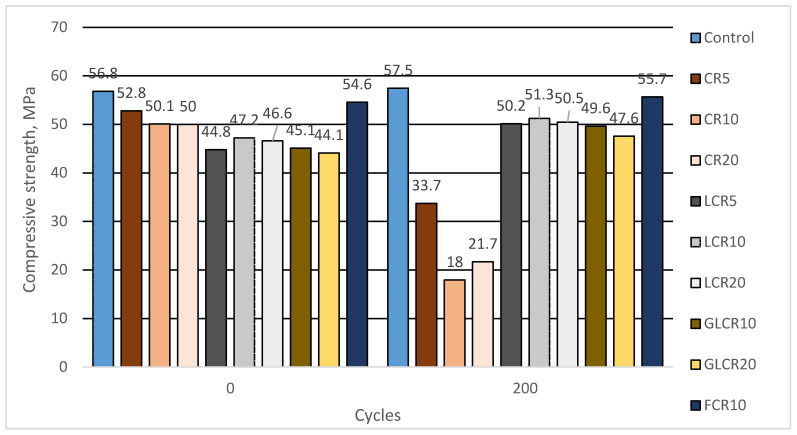
Changes in compressive strength of concrete in freeze-thaw resistance test.

**Figure 6 materials-14-02321-f006:**
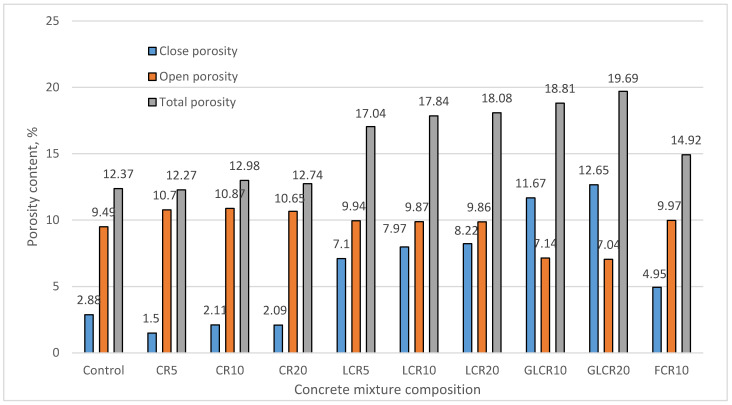
Change in porosity parameters of concrete after water absorption kinetics testing.

**Figure 7 materials-14-02321-f007:**
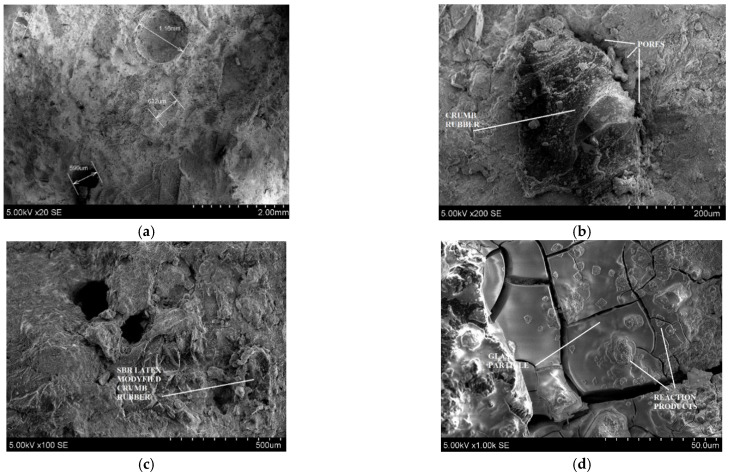
SEM images of concrete particles of prefabricated air bubble (**a**), crumb rubber (**b**), SBR latex modified crumb rubber (**c**), glass powder (**d**), fine crumb rubber (**e**).

**Table 1 materials-14-02321-t001:** Chemical composition of portland cement and glass powder.

Components	Quantity, %	
	CEM I 42.5 R	Glass Powder
SiO_2_	21.01	72.76
TiO_2_	-	0.04
Al_2_O_3_	5.39	1.67
Fe_2_O_3_	3.23	0.79
CaO	62.11	9.74
MgO	1.98	2.09
MnO	-	0.02
Na_2_O	0.38	12.56
K_2_O	0.82	0.76
P_2_O_5_	-	0.02
SO_3_	3.1	0.1
Na_2_Oeq	0.92	13.06
Loss on ignition (%)	2.38	1

**Table 2 materials-14-02321-t002:** Proportions of concrete mixes.

Notation	Compositions for 1 m^3^ of Concrete Mix											
	CR Characteristics		SBR Latex, kg	Cement, kg	Glass Powder, kg	Crushed Granite 4/16, kg	Sand 0/4, kg	Fiber kg	Water, l	Admixtures, wt% of Cement		
										SikaAer Solid	Super-Plastici-zer	Shrinkage Reducer
	Fine-ness	CR, kg										
Control	-	-	-	360	-	999	922	3.5	152.3	2.0	0.8	2.0
CR5	CCR	5					909		152.3	-		
CR10		10					896		152.3			
CR20		20					870		152.3			
LCR5		5	30				909		137.3			
LCR10		10					896		137.3			
LCR20		20					870		137.3			
GLCR10		10		350	10		896		137.3			
GLCR20		20		340	20		870		137.3			
FCR10	FCR	10	-	360	-		896		152.3			

**Table 3 materials-14-02321-t003:** Fresh concrete properties.

Notation	Slump, mm	Density, kg/m^3^	Air Content, %
Control	195	2450	2.5
CR5	230	2400	2.4
CR10	200	2380	3.2
CR20	190	2350	3.4
LCR5	260	2354	5.5
LCR10	260	2321	6.0
LCR20	250	2290	6.8
GLCR10	255	2285	7.2
GLCR20	265	2281	7.5
FCR10	185	2361	2.3

**Table 4 materials-14-02321-t004:** Work and fracture energy used to break the specimens.

Notation	Area, N-m	Fracture Energy, N/m	Residual Flexural Strength at 0.5 mm, MPa	Residual Flexural Strength at 3.5 mm, MPa
Control	8.75	973	2.4	2.55
CR5	10.99	1222	6.64	2.34
CR10	10.45	1161	2.94	3.01
CR20	7.95	883	2.76	2.06
LCR5	5.75	639	1.61	1.66
LCR10	5.15	573	1.8	1.33
LCR20	4.21	468	1.24	1.19
GLCR10	5.32	591	1.55	1.51
GLCR20	7.98	887	2.29	2.28
FCR10	8.58	954	2.75	2.58

**Table 5 materials-14-02321-t005:** Durability parameters of hardened concrete.

Notation	Water Absorption, %	Concrete Density, kg/m^3^	Kf	Predicted Cycles	The Change of Compressive Strength, % Compared to Initial Compressive Strength (Before Freeze-Thaw Test) after 200 Cycles
Control	4.03	2351	3.62	581	+1.19
CR5	4.11	2345	1.55	208	−36.14
CR10	3.72	2311	2.16	330	−64.17
CR20	4.11	2301	2.18	335	−56.62
LCR5	4.46	2231	7.93	>800	+11.98
LCR10	4.47	2210	8.97	>800	+8.60
LCR20	4.48	2203	9.26	>800	+8.33
GLCR10	3.27	2183	18.16	>800	+10.04
GLCR20	3.26	2160	19.97	>800	+7.87
FCR10	4.35	2322	5.52	>800	+1.97

## Data Availability

The data presented in this study are available on request from the corresponding author.

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
