# Peer review of "Mechanical Properties and Durability of Rubberized and Glass Powder Modified Rubberized Concrete for Whitetopping Structures"

_materials, 2021, doi:10.3390/ma14092321_

Round 1

Reviewer 1 Report

This paper is interesting in content and results. Anyway I propose some changes that, in my opinion, could improve the manuscript.

English revision, i.e:

"A high resistance to freezing and thawing (as air entrained agent usually is used prefabricated air burbles because of floated concrete surfaces [3])”

This phrase is not very clear, could be redacted better.

Also this paragraph:

“The end of life vehicle directive 2000/53/EC introduced three legislation to improve tyre manage-ment's use: extended producer responsibility, a tax system, and the free market system. According to this legislation, 100% of waste rubber tyres are reused all over European countries, and disposal rates also started reduced.

There are  others that could be improved. I would suggest general revision of the language, some of the expressions are difficult to understand .

Some of the data provided in thetintroduction could be reviewed otr better explained, In this case, 100% of the tyres are reused? I guess this includes incineration.  

Some of the affirmations related to the chemistry of the interactions of the rubber with the concrete or specifically ,water, are not completely rigorous. Authors consider tires hydrophobic because:

“Rubber is hydrophobic due to zinc stearate (zinc stearate is one of the tyre manufac-turing product). During hydration, the zinc stearate is forming a soap layer on rubber that repelling wàter”

The existence of zinc stearate on the surface of tires was demonstrated in:

The Use of Waxes and Wetting Additives to Improve Compatibility Between HDPE and Ground Tyre Rubber

  1. Cañavate, F. Carrillo, P. Casas, ...2009

https://doi.org/10.1177/0021998309351602

The exudation of zinc stearate, was observed more specifically when the rubber was submitted to a mixing process at high temperature, not at room temperature. Anyway, the existance of zinc stearate on the surface, even being an important factor, is not the only one that may produce hydrophobicity in tires. There are many others including the own nature of the material.  In the reference included in the manuscript,  (20) authors propose a general change of the surface of the rubber eliminating zinc estearate but also producing voids in order to improve wettability.

The affirmations on hydrophobicity become important also later in the manuscript where authors relate this feature to the ability of the rubber to entrap air: Conclusion 5.

I would suggest a deeper discussion about this subject in the paper, since is a topic that generates doubt in many concrete-rubber studies.  Some more information about the nature of the surface of the tires can be also found in:

Composites reinforced with reused tyres: Surface oxidant treatment to improve the interfacial compatibility

 January 2007Composites Part A Applied Science and Manufacturing 38(1):44-50

DOI: 10.1016/j.compositesa.2006.01.022

In this paper authors treated tire rubber with oxidant reactives in order to create polar groups on the surface wich would improve the compatibility with other materials. These papers are only examples that came to my mind, surely some other interesting results will be also available elsewhere.

Figure 4 could be improved, it is small and not very clear

Author Response

Dear Reviewer,

Many thanks for logical and fair comments. I have tried to take your comments into account as much as possible. I hope my answers will match your expectations. In attached word document you will find my response to your notices.

Reviewer 2 Report

Manuscript: “Mechanical properties and durability of rubberized and glass powder modified rubberized concrete for Whitetopping structures”

The manuscript presents interesting information on mechanical and physical properties of rubberized concrete. In general the format of the article is good and the presentation is concise. 

However I can not proceed with a detailed review since the submitted article does not meet the journal requirements.

Most importantly I would suggest to resubmit using the materials journal template (including line numbering). You can download the template here: Microsoft Word template

Furthermore, I would suggest to edit all Figures containing grayscale patterns to differentiate between data series. Replace the patterns with different colors.

Don't use numbering in the Conclusions but rather bullets. Also please decide on the tense you will use. You mix simple past and present in the conclusions.

Finally I would suggest to carefully proofread the manuscript, since I have noticed several grammatical and syntactic errors.

Author Response

Dear Reviewer,

Many thanks for your comments. I hope my answers will mach your expectations. In attached word document you will find my response to your notice.

Reviewer 3 Report

From the analysis of the information presented in the article, I found the following:

- The paper presents a series of results that could be of interest to the scientific community:

- The introductory section should be substantially improved, as reference should be made to research in the field of modified cements with additions of recycled rubber particles, respectively glass particles, and not a presentation of the distribution of rubber waste in tires, respectively glass;

- The Materials section must be substantially developed in the sense that the properties of the rubber particles must be presented;

- The methodology and equipment used to obtain the data that formed the basis of the graph in Figure 1 and the data presented in Table 1 must be specified;

- The admixtures properties presented in Table 2 must be presented;

- The research methodology should be presented more clearly in the sense that it is necessary to present a justification for the decision to choose the proportions of the materials presented in Table 2;

It is necessary to add macroscopic images of the representative obtained samples;

- The parameters that were used to perform the freeze-thaw cycles must be presented;

- The discussion part needs to be improved in order to better highlight the novelty brought by the research presented in the paper compared to other research in the field. In the current form in the discussion part, references are made to a series of bibliographic sources through which the obtained results are confirmed, but the novelty brought by the research cannot be identified;

- In the final part of the conclusions, the future research directions must be presented. The practical applications of the research could also be presented in conclusions.

Author Response

Dear Reviewer,

Many thanks for the logical and fair comments. I have tried to take your comments into account as much as possible. I hope my answers will mach your expectations. In attached word file you will find my response to your notice.

Round 2

Reviewer 2 Report

As I mentioned in my original review, the manuscript does not meet the journal requirements.

Most importantly I would suggest to resubmit using the materials journal template (including line numbering).

You can download the template here: https://www.mdpi.com/files/word-templates/materials-template.dot

Author Response

Now completely was worked on given template word file. Sorry for inconvenience and I hope so now all will be fine

Reviewer 3 Report

Advances in Metallurgical and Material Engineering

Author Response

Thanks for the notice but we believe that our manuscript better fits to construction and building material section. Next time we will think and about this section